materials science/electromagnetism

MoSi$_2$ (high-temperature ceramic), multi-polarization resonance, microwaves absorption

**Author for correspondence:**
Ziming Xiong
e-mail: xzm992311@163.com

This article has been edited by the Royal Society of Chemistry, including the commissioning, peer review process and editorial aspects up to the point of acceptance.

# C,N-codoped MoSi$_2$ ceramic with excellent heat resistance for microwaves absorption application

## Zhiqian Yang, Chang Xu, Yilu Xia and Ziming Xiong

State Key Laboratory for Disaster Prevention and Mitigation of Explosion and Impact, Army Engineering University of PLA, Nanjing 210007, People's Republic of China

 ZY, 0000-0002-3132-1777

Microwave absorption (MA) materials with high heat resistance have a wide range of applications in many fields. In this work, a C,N-codoped MoSi$_2$ ceramic was prepared via a facile solid-phase reaction method and its MA properties was investigated. On the one hand, the results indicate that this ceramic possesses excellent heat resistance and the weight of the MoSi$_2$ is almost constant when the temperature is lower than 800°C. On the other hand, this ceramic shows good MA performance when the filler loading ratio increases to 30 vol%, the value of reflection loss (RL) could reach to −17.70 dB at 7.44 GHz with the thickness of 2.0 mm and the effective electromagnetic absorption bandwidth (RL below −10 dB) could reach to 1.88 GHz (9.28–11.16 GHz) with the thickness of 1.5 mm. Multi-polarization resonance loss is considered as the predominant attention mechanism on the MA performance of this MoSi$_2$ ceramic. This research provides a new idea for understanding resonance mechanism and greatly expands the application scope of MoSi$_2$ ceramic in MA area.

## 1. Introduction

Nowadays, the overdevelopment of electronics industry leads to serious electromagnetic radiation, which is mainly caused by the interference effects induced by electric and magnetic fields emanating from wide range of electrical circuitry [1,2]. Electromagnetic radiation is not only harmful to human health, but also to some sophisticated instruments. Hence, the research on microwave absorption (MA) materials gradually attracts people's attention. At present, many kinds of materials are confirmed to possess great MA performance, such as carbon materials [3–5], conductive polymers [6,7] and chiral materials [8]. However, on the one hand, temperature is one of the most important factors affecting dielectric properties [9–11]; these materials' MA performance varies

greatly at different temperatures. On the other hand, these materials are not stable at exceedingly high temperature and thus restricted to extremely harsh conditions in certain applications.

In recent years, due to high temperature resistance, ultrahigh-temperature ceramic materials have gained great attention. Zhang *et al.* found that $ZrB_2$ ceramic can resist high temperature and have proper reflection loss to electromagnetic wave [12]. Dou *et al.* studied that dielectric properties of N-doped SiC and explained the mechanism of polarization relaxation [13]. Besides, the research of SiC ceramic [14–16] and composite ceramics [17–20] in the field of microwave absorption has been more in-depth. Moreover, ultrahigh-temperature ceramic not only has great mechanical properties and corrosion resistance, but also has great MA properties. Owing to great comprehensive performance, ultrahigh-temperature ceramic has great potential in military and aerospace applications. As an ultrahigh-temperature ceramic, $MoSi_2$ possesses high thermal conductivity (25 W m$^{-1}$ K), moderate density (6.24 g cm$^{-3}$), high melting point (2300 K), relatively lower coefficient of thermal expansion ($8.6 \times 10^{-6}$ K$^{-1}$) and elevated oxidation resistance (about 1600°C) [21]. Besides, $MoSi_2$ has amazing electrical conductivity, which suggests the potential in MA materials. To the best of our knowledge, however, the MA properties of $MoSi_2$ remain to be explored.

In this study, we report the MA properties of a C,N-codoped $MoSi_2$ ceramic, which was prepared from a solid-state reaction method [22]. The MA performance and dielectric properties of $MoSi_2$ in the frequency range of 2–18 GHz were investigated in detail. The results indicate that C,N-codoped $MoSi_2$ ceramic shows excellent heat resistance as well as good MA performance. A multi-polarization resonance loss model was used to investigated the attention mechanism of C,N-codoped $MoSi_2$ ceramic toward MA.

# 2. Material and methods

## 2.1. Material

The anhydrous ethanol was purchased from Shanghai GENERAL-REAGENT Titan Scientific Co., Ltd, China. The $MoO_2$ was purchased from Shanghai Yunfu Nanotechnology Co., Ltd, China. The Si was purchased from Beijing Hongyu New Material Technology Co., Ltd, China.

## 2.2. Synthesis of C,N-codoped MoSi$_2$ ceramic

$MoSi_2$ was constructed according to the following steps. $MoO_2$ (36.0 g) and Si (24.0 g) were added to the jar mill in turn. Next, ethanol (2.0 ml) was added to the jar mill. Then, the above mixture was ground for 8 h. Next, the mixture was heated for 8 h at temperature of 80°C. After drying, the mixture was passed through a 60 mesh screen and put into a graphite crucible. Then, the graphite crucible was put into pressure-free sintering furnace that is full of nitrogen, the material was sintered at heating rate of 10°C min$^{-1}$ from room temperature to 1400°C and kept at 1400°C for 1 h.

## 2.3. Characterization

The micro morphology of $MoSi_2$ samples were characterized by scanning electron microscope (SEM, Hitachi, S4800) and transmission electron microscope (TEM; FEI, Tecnai G2 F20). The crystal feature of $MoSi_2$ samples were investigated by instrument (XRD; Philips, X' Pert Pro) by Cu K$\alpha$ ($\lambda = 1.54$ Å) radiation source (30.0 mA, 40.0 kV). The result of thermostability analysis was evaluated by thermo gravimetric analyser (TGA; TA, SDT Q600) in the air. The result of X-ray photoelectron spectroscopy (XPS) was got from an apparatus (ESCALABTM 250Xi, Thermo Fisher Scientific). $MoSi_2$/wax was prepared by uniformly mixing $MoSi_2$ in wax and pressed into a toroidal-shaped specimen (Φin: 3.0 mm, Φout: 7.03 mm). The electromagnetic properties of $MoSi_2$ were recorded using a network analyser (VNA; N5242A PNA-X, Agilent).

# 3. Results and discussion

## 3.1. Morphology and microstructure

Figure 1 shows a typical morphology of $MoSi_2$. Figure 1*a–d* displays SEM images of $MoSi_2$ at different magnifications, and a lot of irregularly shaped block structure can be observed. The block size ranges from 0.5 to 8 μm. The elemental mappings of $MoSi_2$ are shown in figure 1*e,f*. Generally, the

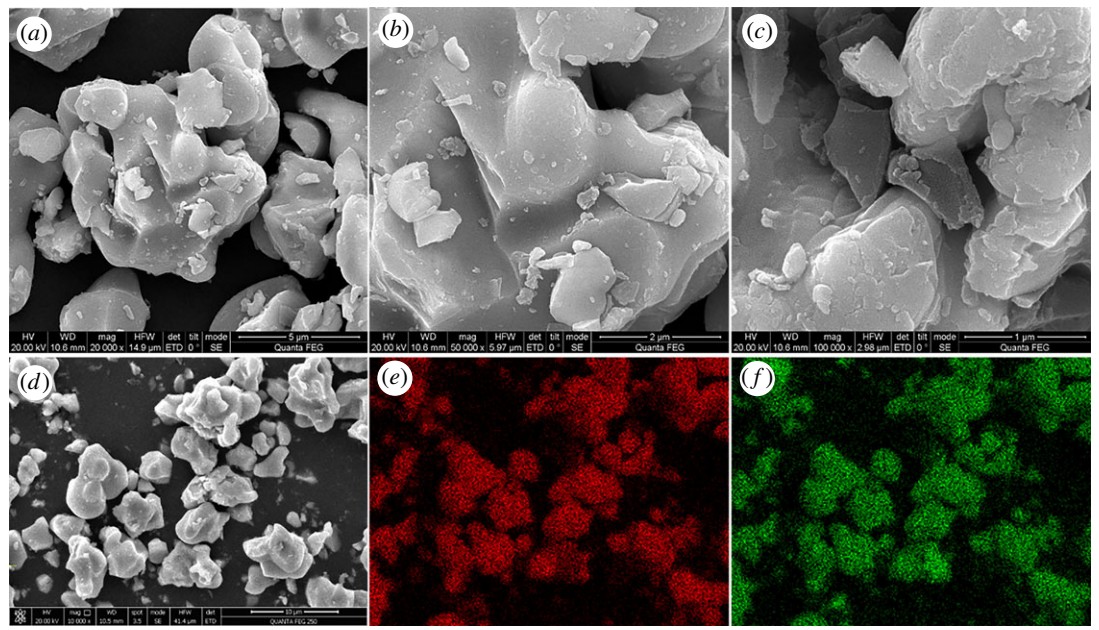

**Figure 1.** (*a*–*d*) Representative SEM image of MoSi₂ ceramic; (*e,f*) representative elemental mapping analysis of MoSi₂ ceramic.

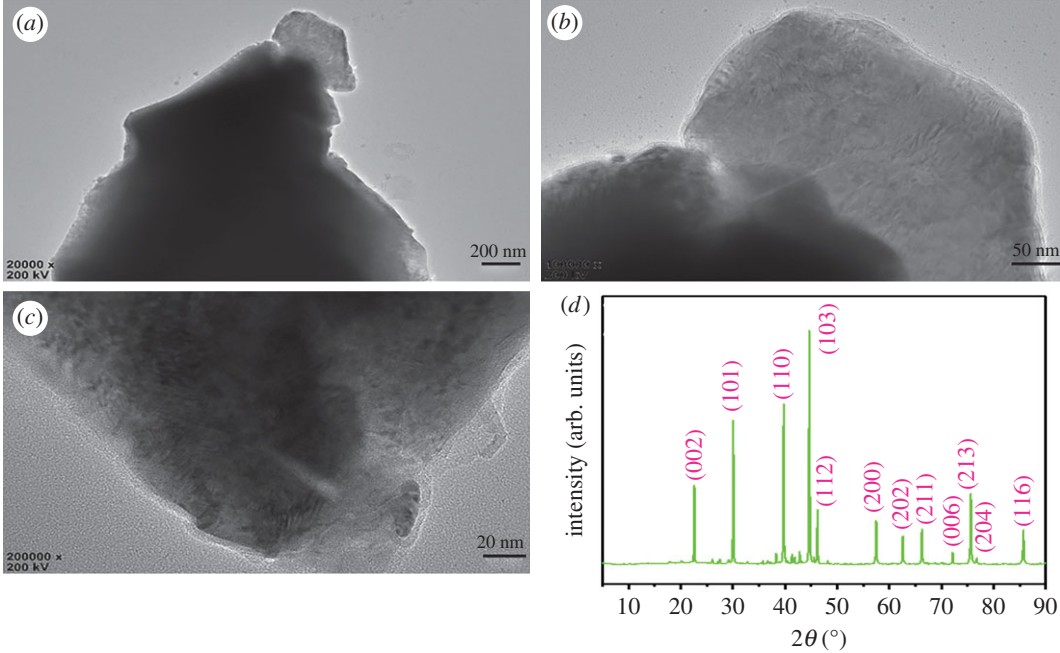

**Figure 2.** (*a*–*c*) Representative TEM image of MoSi₂ ceramic; (*d*) XRD patterns of MoSi₂ ceramic.

atomic ratio of Mo/Si was about 1 : 2. However, it is suggested that the signal of Si was a little stronger than that of Mo from figure 1*e,f*, which suggests the ratio of Mo/Si is lower than 1 : 2. Besides, it could be found that both Mo and Si were uniformly distributed.

Figure 2*a*–*c* shows the inner structure of MoSi₂ ceramics, which displays the irregularly shaped block structure was composed of layered structure by means of stacking. The XRD pattern of MoSi₂ ceramic is exhibited in figure 2*d*. The diffraction peaks at 22.6, 30.1, 39.7, 44.6, 46.2, 57.4, 62.5, 66.2, 72.1, 75.5, 76.7 and 85.6° matched well to the (002), (101), (110), (103), (112), (200), (202), (211), (006), (213), (204) and (116) planes of MoSi₂ (PDF#41–0612), which suggest MoSi₂ is polycrystalline.

## 3.2. Elemental composition and thermal stability of MoSi₂

In order to further investigate the element composition and structure of MoSi₂ ceramic, XPS was performed and is shown in figure 3*a*–*d* The peaks of C, N, O, Mo and Si can be observed in the full spectrum in

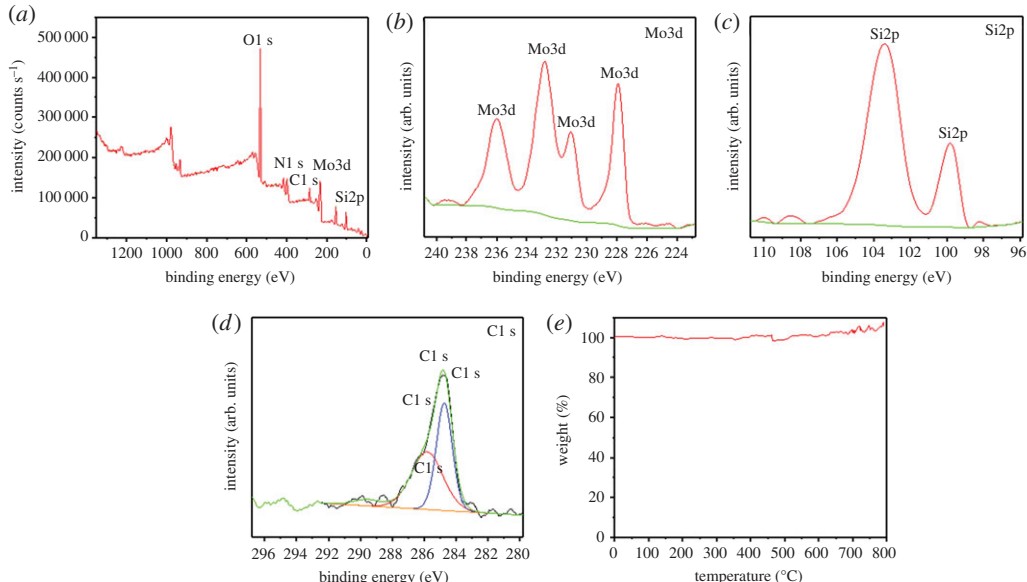

**Figure 3.** (*a–d*) XPS spectra of MoSi₂ ceramic, (*e*) TGA of MoSi₂ ceramic.

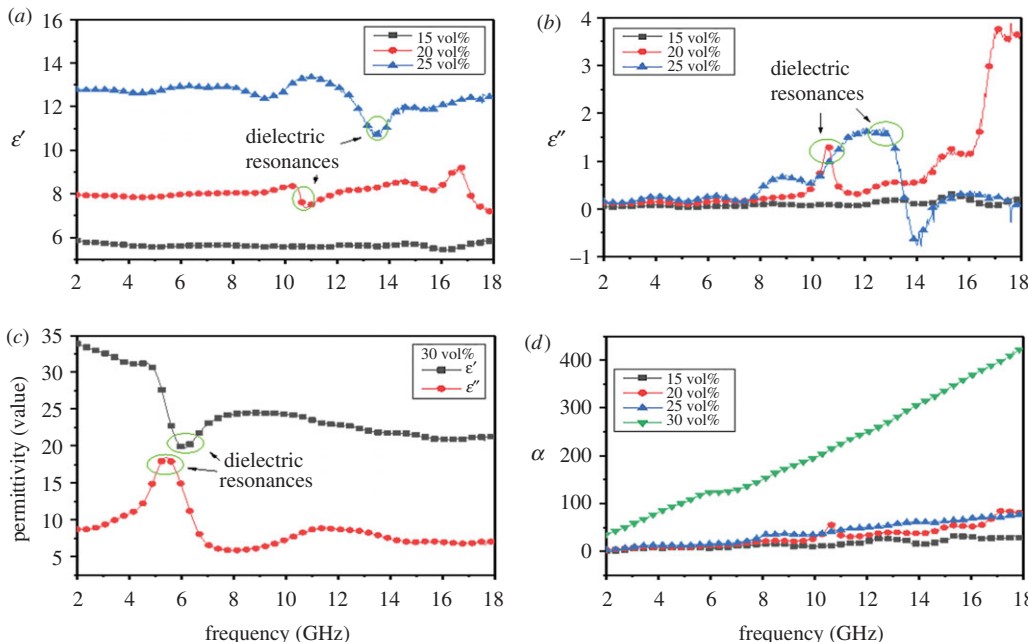

**Figure 4.** (*a–c*) Frequency and filler loading ratio dependence of complex permittivity; (*d*) attenuation constant ($\alpha$) of MoSi₂ ceramic with different filler loading ratios.

figure 3*a*, the existence of O mainly due to formation of small amounts of SiO₂. Figure 3*b,c* are XPS spectra of Mo and Si. Mo3d peaks are located in the range of binding energies of 236 to 227 eV and Si2p peaks are located in the range of binding energies of 104 to 99 eV. The peak of C1 s is located at binding energies of 284 eV according to figure 3*d*. The atomic ratio of Mo and Si is 21.06 : 78.94, which is close to 1 : 4. The ratio is higher than 1 : 2. It proves that the sample consists of a small amount of SiO₂ and excessive Si. Figure 3*e* shows the thermal stability of MoSi₂ ceramic; the weight of MoSi₂ is almost constant when the temperature is below 800°C, which suggests that MoSi₂ is an ultrahigh-temperaturee ceramic material.

## 3.3. Dielectric and microwave absorption properties

Figure 4*a–c* shows the real part ($\varepsilon'$) and the imaginary part ($\varepsilon''$) of the relative permittivity ($\varepsilon_r$) of MoSi₂ ceramic with different filler loading ratios. Generally speaking, the electromagnetic storage ability is

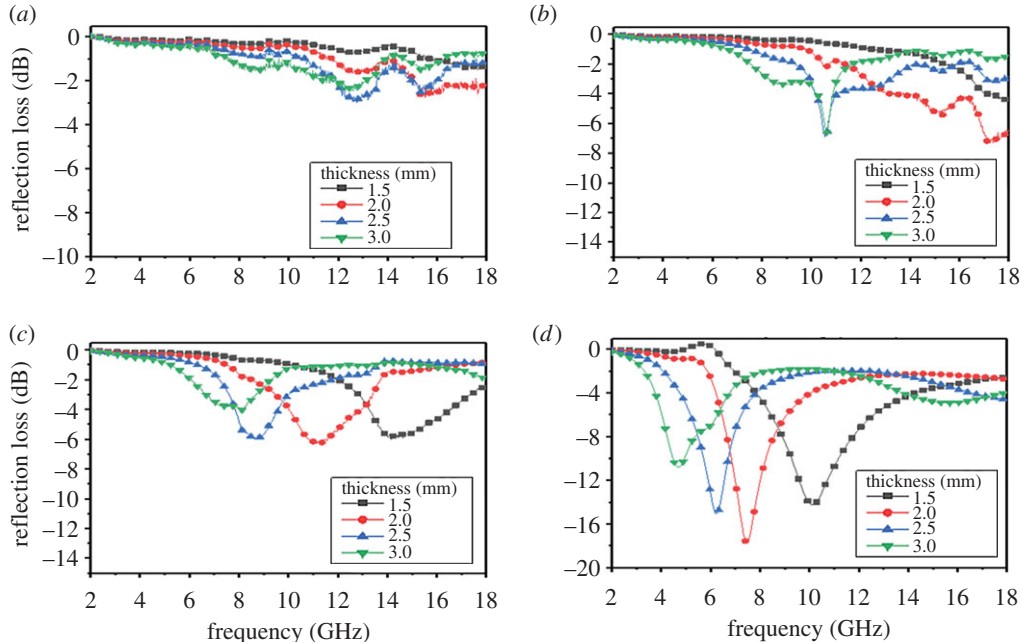

**Figure 5.** RL curves of MoSi$_2$ ceramic with filler loading ratios of (*a*) 15 vol%, (*b*) 20 vol%, (*c*) 25 vol% and (*d*) 30 vol%.

subject to the real part ($\varepsilon'$) of complex permittivity while the energy damping is determined by the imaginary part ($\varepsilon''$) of complex permittivity [23]. As the filler loading ratio increasing, both $\varepsilon'$ and $\varepsilon''$ increase. When the filler loading ratio is 30 vol%, dielectric parameter of MoSi$_2$ increases a lot. According to the Debye theory, the frequency is negatively correlated with the permittivity of a composite [24]. When the frequency is at 12.72, 10.56 and 5.52 GHz, resonance peaks can be observed from figure 4*a*,*b* and the value of $\varepsilon'$ at peak point is greatly improved. Hence, the loss mechanism of MoSi$_2$ includes polarization loss and resonance loss. Around the 12.72 GHz of resonance peaks, the $\varepsilon''$ of 25 vol% addition of MoSi$_2$ drops to a negative value, which suggests this material has potential to be a metamaterial. Besides, resonance peaks shift to lower frequency and become higher when the filler loading ratio increases. Figure 4*c* shows that the resonance signal is much stronger when the addition amount is 30 vol%. As shown in figure 4*d*, attenuation constant ($\alpha$) of MoSi$_2$ ceramic is enhanced with the increase of the filler loading ratio, especially the filler loading ratio increases from 20 to 30 vol%, which indicates the energy attenuation performance of MoSi$_2$ is improved as the filler loading ratio increase.

The reflection loss (RL) can directly reflect the microwave-absorbing capacity of materials, which is given as [25]

$$R_L = 20\log\frac{|Z_{in} - 1|}{|Z_{in} + 1|}. \tag{3.1}$$

Here, the normalized input impedance $Z_{in}$ of microwave absorption layer is

$$Z_{in} = Z_0\sqrt{\frac{\mu_r}{\varepsilon_r}}tanh\left[j\frac{2\pi}{c}\sqrt{\mu_r\varepsilon_r}fd\right], \tag{3.2}$$

where $Z_0$ is free space impedance, complex permittivity ($\varepsilon_r$) can be expressed as $\varepsilon_r = \varepsilon' - \varepsilon''$, complex permeability ($\mu_r$) can be expressed as $\mu_r = 1$, because the samples are non-magnetic. $f$ is frequency, $d$ is sample thickness and $c$ is light speed.

Figure 5 shows the RL curves of MoSi$_2$ ceramic coupled with different filler loadings and various thicknesses at the frequency range of 2–18 GHz. The absorption peaks are mainly around 12.72, 10.56 and 5.52 GHz, which attributes to the polarization resonance loss (figure 5*a*–*d*). As the filler loading ratio increases, the MA performance of MoSi$_2$ is enhanced. Meanwhile, the absorption peaks shift to lower frequency, which corresponds to permittivity. This further explains that the enhancement of MA performance is attributed to polarization resonance loss. When the filler loading ratio is lower than 25 vol%, the MA performance of MoSi$_2$ ceramic is low. This is owing to the fact that the amount of MoSi$_2$ is not enough to form a conductive network, which leads to a low dielectric loss.

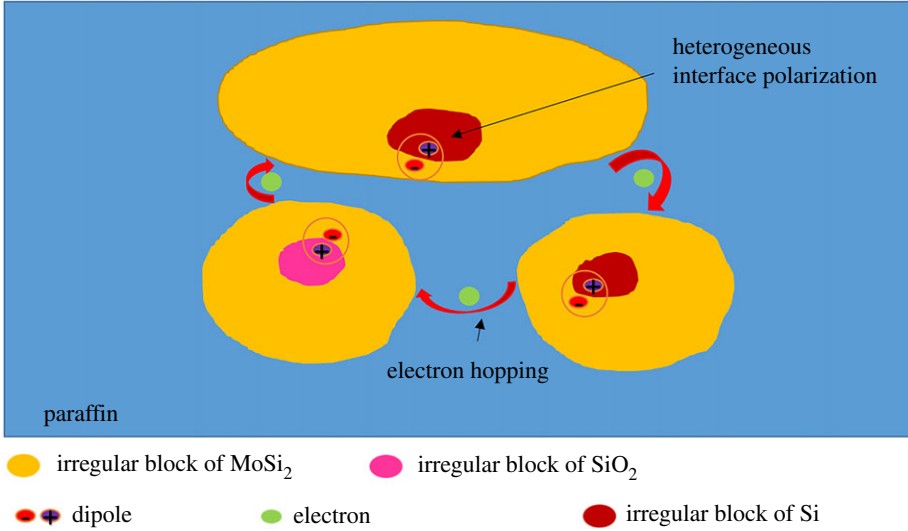

**Figure 6.** Interface effect and conductive network between $SiO_2$, Si and $MoSi_2$.

However, when the filler loading ratio increases to 30 vol%, the value of RL could reach to −17.70 dB at 7.44 GHz with the thickness of 2.0 mm and the effective electromagnetic absorption bandwidth (RL below −10 dB) can reach to 1.88 GHz (9.28–11.16 GHz) with the thickness of 1.5 mm. The MA performance of $MoSi_2$ is enhanced 250% compared to the $MoSi_2$ with addition lower than 25 vol%. Besides, the different thickness corresponds to different absorption peaks. When the thickness becomes thinner, the absorption peaks may be shift to the low frequency. Generally, the reason for the fact that MA performance of $MoSi_2$ is enhanced is the multi-polarization resonance phenomenon. This phenomenon has been proven in permittivity and reflection loss.

## 3.4. Multi-polarization resonance

In general, the MA performance is related to multi-polarization resonance loss, which explains the enhanced MA performance of $MoSi_2$. Figure 6 shows interface effect and conductive network between $SiO_2$, Si and $MoSi_2$. The heterogeneous interfacial polarization in $SiO_2$, Si and $MoSi_2$ is regarded as the capacitor-like structure, which is effective in the MA [26]. Besides, the electron could leap on the irregular block of $MoSi_2$ after absorbing energy, which leads to the formation of conductive network of each block [26]. The existence of conductive network greatly improves the conductivity of $MoSi_2$, which is attributed to the increase of the filler loading ratio. The existence of heterogeneous interface may lead to resonance loss. Resonance loss results in the MA performance of $MoSi_2$ greatly improved at a certain frequency.

## 4. Conclusion

In summary, C,N-codoped $MoSi_2$ ceramic was prepared by solid-state reaction method. The morphology, dielectric properties and MA performance of $MoSi_2$ was investigated. The result of SEM and TEM indicate that the morphology of $MoSi_2$ ceramic is irregular block structure. XPS reveals that C, N and O exist in $MoSi_2$ ceramic. The curve of TGA suggests that $MoSi_2$ has great thermal stability. Moreover, the result suggests that multi-polarization resonance leads to the enhanced MA performance of $MoSi_2$ ceramic. As the filler loading increases, resonance signal is enhanced and resonance peak shifts to the lower frequency. Besides, the MA performance of $MoSi_2$ is greatly improved at the resonance peak. When the filler loading ratio is 30 vol%, the value of RL could reach to −17.70 dB at 7.44 GHz with the thickness of 2.0 mm and the effective electromagnetic absorption bandwidth (RL below −10 dB) can reach to 1.88 GHz (9.28–11.16 GHz) with the thickness of 1.5 mm, which indicates multi-polarization resonance mechanism contributes to improvement of the MA performance of ceramic. The mechanism of multi-polarization resonance could be explained by the existence of heterogeneous interfaces in the material structure, which has been proven in XPS. This study reveals the MA performance of $MoSi_2$ ceramic and suggests ultrahigh-temperature ceramic has potential for MA field.

**Ethics.** We were not required to complete an ethical assessment prior to conducting our research.

**Permission to carry out fieldwork.** No permissions were required prior to conducting our research.

**Data accessibility.** The raw data of TGA, XPS, XRD and electromagnetic parameters supporting this article are available at the Dryad Digital Repository: https://doi.org/10.5061/dryad.vdncjsxr7 [27].

**Authors' contributions.** Z.Y. completed the experimental work and manuscript. C.X. explained the mechanism of multi-polarization resonance. Y.X. and Z.X. interpreted the results and revised the manuscript. All authors contributed to the manuscript and approved the final version.

**Competing interests.** We declare we have no competing interests.

**Funding.** This work was financially supported by the China Postdoctoral Science Foundation (grant no. 2018M640488).

**Acknowledgements.** We thank the editor and anonymous reviewers for their valuable comments and suggestions.

**Disclaimer.** We have read the information for authors and publish policy. We all agree with the policy and prepare the manuscript in accordance with the guidance

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
