## [Reviewer comments · Royal Society Open Science]

Review History

RSOS-200740.R0 (Original submission)

Review form: Reviewer 1

Is the manuscript scientifically sound in its present form?

Yes

Are the interpretations and conclusions justified by the results?

Yes

Is the language acceptable?

Yes

Do you have any ethical concerns with this paper?

Yes

Have you any concerns about statistical analyses in this paper?

Yes

Recommendation?

Accept with minor revision (please list in comments)

Comments to the Author(s)

This manuscript (ID=RSOS-200740) started a study about C,N-codoped-MoSi₂ ceramic was prepared via a facile solid phase reaction method, and its MA properties was investigated. The material can resist high temperature and have proper reflection loss to electromagnetic wave. Comparatively speaking, the composites have potential to be a stable material to weaken the microwave strength. Overall, the manuscript has quite clear layout and is well written. In consequence, I recommend this manuscript for publication in Royal Society Open Science after some minor revisions.

1. The formula of reflection loss should contain magnetic parameter μ , so for non-magnetic samples, it is also necessary to state that $\mu = 1$.
2. There are some grammar mistakes and mis-spellings, please check the whole manuscript.
3. On introduction section, the authors should summarize similar reports about materials of microwave absorption to stress their novelty.
4. Theoretically, the value of ϵ should not be negative, for most occasions it might be the error of vector network analyzer. However, it is mentioned in the manuscript that 25 vol% sample has a potential to be a metamaterial. Why you get this conclusion, as other samples do not have this characteristic? Please give the mechanism explanation.

Review form: Reviewer 2

Is the manuscript scientifically sound in its present form?

Yes

Are the interpretations and conclusions justified by the results?

Yes

Is the language acceptable?

Yes

Do you have any ethical concerns with this paper?

No

Have you any concerns about statistical analyses in this paper?

No

Recommendation?

Major revision is needed (please make suggestions in comments)

Comments to the Author(s)

In this paper, the preparation and properties of C,N-Codoped MoSi₂ Ceramic have been studied in detail, and the MA and dielectric properties of MoSi₂ Ceramics have been explored. This work is well structured and easy to read, the research objectives and methods are sound, in line with the journal scope of RSOS. The following minor comments might help to improve the quality of the manuscript:

- 1) 3.2 Synthesis of C,N-codoped MoSi₂ Ceramic: "the mixture was heated for 8h at temperature of 80°C", "the material was sintered at heating rate of 10°C • min⁻¹ from room temperature to 1400°C". Please explain the reasons for choosing 80°C, 1400°C and the heating rate. Why not use sectional heating?
- 2) 4.2 Elemental Composition and Thermal Stability of MoSi₂: Compared with pure MoSi₂, which element in C,N-Codoped MoSi₂ has the greatest influence on its thermal stability? Why doping with C and N? What are the advantages of C and N?
- 3) 4.3 Dielectric and MA Properties: Temperature has a particularly important effect on the dielectric properties. Therefore, the authors are encouraged to add a short discussion about the

relation of dielectric properties of sample and temperature. There are some recent publications recommended to add in the reference section:

- i. Omran M , Li K , Chen J , et al. Dielectric properties and thermal behavior of electrolytic manganese anode mud in microwave field[J]. Journal of Hazardous Materials, 2019, 384:121227.
 - ii. Li K, Chen G, Li X, et al. High-temperature dielectric properties and pyrolysis reduction characteristics of different biomass-pyrolusite mixtures in microwave field[J]. Bioresource Technology, 2019.
 - iii. Guo Chen, Kangqiang Li, Mamdouh Omran, et al. Investigations on the microwave absorption properties and thermal behavior of vanadium slag: Improvement in microwave oxidation roasting for recycling vanadium and chromium. Journal of Hazardous Materials, 2020.
- 4) Some sentences are grammatically wrong. Please check the word case, punctuation, tense and other issues.

I hope these suggestions could be well accepted by the authors.

Decision letter (RSOS-200740.R0)

Dear Dr Yang:

Title: C,N-Codoped MoSi₂ Ceramic with Excellent Heat Resistance for Microwaves Absorption Application
Manuscript ID: RSOS-200740

The editor assigned to your manuscript has now received comments from reviewers. We would like you to revise your paper in accordance with the referee and Subject Editor suggestions which can be found below (not including confidential reports to the Editor). Please note this decision does not guarantee eventual acceptance.

Please submit your revised paper before 21-Jun-2020. Please note that the revision deadline will expire at 00.00am on this date. If we do not hear from you within this time then it will be assumed that the paper has been withdrawn. In exceptional circumstances, extensions may be possible if agreed with the Editorial Office in advance. We do not allow multiple rounds of revision so we urge you to make every effort to fully address all of the comments at this stage. If deemed necessary by the Editors, your manuscript will be sent back to one or more of the original reviewers for assessment. If the original reviewers are not available we may invite new reviewers.

On behalf of the Subject Editor Professor Anthony Stace and the Associate Editor Dr Chaohua Cui.

RSC Associate Editor:
Comments to the Author:
(There are no comments.)

RSC Subject Editor:
Comments to the Author:
(There are no comments.)

Reviewers' Comments to Author:
Reviewer: 1

Comments to the Author(s)

This manuscript (ID=RSOS-200740) started a study about C,N-codoped-MoSi₂ ceramic was prepared via a facile solid phase reaction method, and its MA properties was investigated. The material can resist high temperature and have proper reflection loss to electromagnetic wave. Comparatively speaking, the composites have potential to be a stable material to weaken the microwave strength. Overall, the manuscript has quite clear layout and is well written. In consequence, I recommend this manuscript for publication in Royal Society Open Science after some minor revisions.

1. The formula of reflection loss should contain magnetic parameter μ , so for non-magnetic samples, it is also necessary to state that $\mu = 1$.
2. There are some grammar mistakes and mis-spellings, please check the whole manuscript.
3. On introduction section, the authors should summarize similar reports about materials of microwave absorption to stress their novelty.
4. Theoretically, the value of ϵ should not be negative, for most occasions it might be the error of vector network analyzer. However, it is mentioned in the manuscript that 25 vol% sample has a potential to be a metamaterial. Why you get this conclusion, as other samples do not have this characteristic? Please give the mechanism explanation.

Reviewer: 2

Comments to the Author(s)

In this paper, the preparation and properties of C,N-Codoped MoSi₂ Ceramic have been studied in detail, and the MA and dielectric properties of MoSi₂ Ceramics have been explored. This work is well structured and easy to read, the research objectives and methods are sound, in line with the journal scope of RSOS. The following minor comments might help to improve the quality of the manuscript:

1) 3.2 Synthesis of C,N-codoped MoSi₂ Ceramic: "the mixture was heated for 8h at temperature of 80°C", "the material was sintered at heating rate of 10°C•min⁻¹ from room temperature to 1400°C". Please explain the reasons for choosing 80°C, 1400°C and the heating rate. Why not use sectional heating?

2) 4.2 Elemental Composition and Thermal Stability of MoSi₂: Compared with pure MoSi₂, which element in C,N-Codoped MoSi₂ has the greatest influence on its thermal stability? Why doping with C and N? What are the advantages of C and N?

3) 4.3 Dielectric and MA Properties: Temperature has a particularly important effect on the dielectric properties. Therefore, the authors are encouraged to add a short discussion about the relation of dielectric properties of sample and temperature. There are some recent publications recommended to add in the reference section:

i. Omran M , Li K , Chen J , et al. Dielectric properties and thermal behavior of electrolytic manganese anode mud in microwave field[J]. Journal of Hazardous Materials, 2019, 384:121227.

ii. Li K, Chen G, Li X, et al. High-temperature dielectric properties and pyrolysis reduction characteristics of different biomass-pyrolusite mixtures in microwave field[J]. Bioresource Technology, 2019.

iii. Guo Chen, Kangqiang Li, Mamdouh Omran, et al. Investigations on the microwave absorption properties and thermal behavior of vanadium slag: Improvement in microwave oxidation roasting for recycling vanadium and chromium. Journal of Hazardous Materials, 2020.

4) Some sentences are grammatically wrong. Please check the word case, punctuation, tense and other issues.

I hope these suggestions could be well accepted by the authors.

Author's Response to Decision Letter for (RSOS-200740.R0)

See Appendix A.

Decision letter (RSOS-200740.R1)

Dear Dr Yang:

Title: C,N-Codoped MoSi₂ Ceramic with Excellent Heat Resistance for Microwaves Absorption Application

Manuscript ID: RSOS-200740.R1

It is a pleasure to accept your manuscript in its current form for publication in Royal Society Open Science. The chemistry content of Royal Society Open Science is published in collaboration with the Royal Society of Chemistry.

On behalf of the Subject Editor Professor Anthony Stace and the Associate Editor Dr Chaohua Cui.

RSC Associate Editor
Comments to the Author:
(There are no comments.)

Reviewer(s)' Comments to Author:

Appendix A

Response to editor and reviewers

Dear Editors and Reviewers:

Thank you for your letter and for the reviewers' comments concerning our manuscript entitled "C,N-Codoped MoSi₂ Ceramic with Excellent Heat Resistance for Microwaves Absorption Application" (ID: RSOS-200740). These comments are all valuable and very helpful for revising and improving our paper, as well as the important guiding significance to our researches. We have studied comments carefully and have made correction which we hope meet with approval.

Reviewer: 1

1. The formula of reflection loss should contain magnetic parameter μ , so for non-magnetic samples, it is also necessary to state that $\mu=1$.

Response: Thank you for the helpful suggestion. We have supplemented the information of μ in the formula. The changes are marked with " " in the revised manuscript and supporting information.

2. There are some grammar mistakes and mis-spellings, please check the whole manuscript.

Response: Thank you for reminding. We have tried our best to check the whole paper carefully and correct the grammar mistakes.

3. On introduction section, the authors should summarize similar reports about materials of microwave absorption to stress their novelty.

Response: Thank you for reminding. More similar reports about materials of microwave absorption was added. The changes are marked with " " in the revised manuscript.

4. Theoretically, the value of ϵ should not be negative, for most occasions it might be the error of vector network analyzer. However, it is mentioned in the manuscript that 25 vol% sample has a potential to be a metamaterial. Why you get this conclusion, as

other samples do not have this characteristic? Please give the mechanism explanation.

Response: Thank you for reminding. We thought the reason why ϵ is a negative value was the error of vector network analyzer at the beginning. However, the repeated experimental results show that MoSi_2 may have the potential to become a metamaterial.

Reviewer: 2

1. Synthesis of C,N-codoped MoSi_2 Ceramic: “the mixture was heated for 8h at temperature of 80°C ”, “the material was sintered at heating rate of $10^\circ\text{C}\cdot\text{min}^{-1}$ from room temperature to 1400°C ”. Please explain the reasons for choosing 80°C , 1400°C and the heating rate. Why not use sectional heating?

Response: Thank you for the helpful suggestion. The reasons for choosing 80°C , 1400°C is that 80°C and 1400°C is two processes in the experiment. In order to make MoO_2 and Si disperse as evenly as possible, ethyl alcohol was used as the solvent and 80°C is to make the solvent evaporate as soon as possible. After evaporation, large particles need to be filtered at the room temperature. According to previous research, the synthesis temperature of MoSi_2 is mainly between 1400 - 1500°C . When the temperature is lower than 1300°C , impurities such as Mo_5Si_3 will be produced. Therefore, choosing 1400°C can reduce the formation of impurities, and the reaction is more complete. The heating rate should not be too low, otherwise the reaction time will be too long. If the heating rate is too high, the reactants will be unevenly heated. Thus, we choose $10^\circ\text{C}\cdot\text{min}^{-1}$ as the heating rate.

2. Elemental Composition and Thermal Stability of MoSi_2 : Compared with pure MoSi_2 , which element in C, N-Codoped MoSi_2 has the greatest influence on its thermal stability? Why doping with C and N? What are the advantages of C and N?

Response: Thank you for reminding. From the TG curve, we can infer that the carbon and nitrogen elements hardly have influence on the thermal stability of MoSi_2 ceramics. These two elements were introduced into the system mainly to improve the MA properties of MoSi_2 ceramics. The increase of C provides numerous conductivity paths

for electrons hopping as well as the enhancement conduction loss. Meanwhile, the existence of amorphous C and N means more defects in the nanostructure which results in generating sufficient interface polarization, resonance relaxation, scatter response, and therefore increased dipole polarization in the nanocomposites.

3. Dielectric and MA Properties: Temperature has a particularly important effect on the dielectric properties. Therefore, the authors are encouraged to add a short discussion about the relation of dielectric properties of sample and temperature. There are some recent publications recommended to add in the reference section:

- i. Omran M, Li K, Chen J, et al. Dielectric properties and thermal behavior of electrolytic manganese anode mud in microwave field[J]. *Journal of Hazardous Materials*, 2019, 384:121227.
- ii. Li K, Chen G, Li X, et al. High-temperature dielectric properties and pyrolysis reduction characteristics of different biomass-pyrolusite mixtures in microwave field[J]. *Bioresource Technology*, 2019.
- iii. Guo Chen, Kangqiang Li, Mamdouh Omran, et al. Investigations on the microwave absorption properties and thermal behavior of vanadium slag: Improvement in microwave oxidation roasting for recycling vanadium and chromium. *Journal of Hazardous Materials*, 2020.

Response: Thank you for the helpful suggestion. This problem is worthy of our deep consideration. Temperature does have a significant effect on MA performance. However, limited by the test platform, we can't complete this part of the test. We have cited the key refs to extend this submission. The changes are marked with “ ” in the revised manuscript.

4. Some sentences are grammatically wrong. Please check the word case, punctuation, tense and other issues.

Response: Thank you for reminding. We have tried our best to check the whole paper carefully and correct the grammar mistakes and punctuation.

We appreciated for editors and reviewers warm work earnestly, and hope that the correction will meet with approval. Once again, thank you very much for your comments and suggestions.

Corresponding author: ZhiQian Yang njlgdyzq@sina.com